

# Self-supervised recurrent depth estimation with attention mechanisms

Ilya Makarov[1,2,3,*], Maria Bakhanova[1,*], Sergey Nikolenko[4,5] and Olga Gerasimova[1]

[1] HSE University, Moscow, Russia
[2] Artificial Intelligence Research Institute (AIRI), Moscow, Russia
[3] Big Data Research Center, National University of Science and Technology MISIS, Moscow, Russia
[4] Steklov Institute of Mathematics at St. Petersburg, St. Petersburg, Russia
[5] St. Petersburg State University, St. Petersburg, Russia
[*] These authors contributed equally to this work.

## ABSTRACT

Depth estimation has been an essential task for many computer vision applications, especially in autonomous driving, where safety is paramount. Depth can be estimated not only with traditional supervised learning but also via a self-supervised approach that relies on camera motion and does not require ground truth depth maps. Recently, major improvements have been introduced to make self-supervised depth prediction more precise. However, most existing approaches still focus on single-frame depth estimation, even in the self-supervised setting. Since most methods can operate with frame sequences, we believe that the quality of current models can be significantly improved with the help of information about previous frames. In this work, we study different ways of integrating recurrent blocks and attention mechanisms into a common self-supervised depth estimation pipeline. We propose a set of modifications that utilize temporal information from previous frames and provide new neural network architectures for monocular depth estimation in a self-supervised manner. Our experiments on the KITTI dataset show that proposed modifications can be an effective tool for exploiting temporal information in a depth prediction pipeline.

## INTRODUCTION

Depth estimation is an essential basic problem for a variety of computer vision applications. It is key to understanding 3D scene geometry and can provide essential cues in many tasks, including object detection (*Gupta et al., 2014*), scene reconstruction (*Menze, Heipke & Geiger, 2015*; *Menze, Heipke & Geiger, 2018*; *Shin et al., 2019*), simultaneous localization and mapping (SLAM) (*Yang et al., 2018*), image classification (*He, 2017*), and others. In practice, depth maps can be obtained directly, *e.g.*, with the help of light detection and ranging (LiDAR) devices. Although LiDARs are widely used in practice, they produce low-resolution results, and their perception distance is short. Modern methods of depth estimation from image sequences do not suffer from these limitations, and due to the growing success and generally excellent performance of modern computer vision models

Corresponding authors
Ilya Makarov, iamakarov@hse.ru
Olga Gerasimova, ogerasimova@hse.ru

and algorithms they can often supplement or even replace hardware sensors. In this work, we consider the problem of depth estimation through joint optimization of scene structure and camera motion across sequences of RGB images. This approach is called *Structure from Motion* (SfM) (*Schnberger & Frahm, 2016*), and it implies training in a self-supervised manner.

Over the past decade, depth estimation has been considered in numerous studies. Many theoretical and conceptual frameworks have been put forward to boost depth prediction quality (*Eigen, Puhrsch & Fergus, 2014*; *Laina et al., 2016b*; *Eigen & Fergus, 2014*; *Fu et al., 2018*). Recent improvements in self-supervised depth estimation have also resulted in important performance improvements (*Zhou et al., 2017*; *Godard et al., 2019*; *Guizilini et al., 2020a*). We especially note methods that combined both approaches: depth prediction and information produced by a LiDAR sensor or SLAM (*Ma & Karaman, 2017*; *Ma, Cavalheiro & Karaman, 2018*).

Despite the progress of modern depth estimation methods, a number of challenging problems still remain. One key problem is the fact that depth estimation is ill-posed: the same input image can project to different depths. Moreover, existing depth estimation models are often very sensitive to changing external conditions such as lighting and noise. Self-supervised approaches to depth estimation rely on ego-motion information and do not take into account ground truth depth maps, which implies that these methods usually perform worse than supervised ones.

Existing approaches, both supervised and self-supervised, usually estimate depth based on a single image, although in practice it is very often possible to utilize information from image sequences. There do exist state-of-the-art self-supervised methods that already use frame sequences for camera pose estimation and view synthesis further down the line (*Zhou et al., 2017*; *Godard et al., 2019*). However, most of them still do not utilize this temporal information in depth prediction. We believe that the performance of current state-of-the-art methods can be further improved with temporal knowledge. To achieve this, we aim to provide new modifications that can potentially increase depth accuracy in the monocular setting from an image sequence.

In this study, we focus on self-supervised depth estimation. We propose a modified architecture that can utilize temporal information in a depth model, integrating recurrent ConvGRU and Fusion blocks with self-attention into existing self-supervised training pipelines. To summarize, our main contributions are as follows:

1. We propose a set of modifications that result in new network architectures that utilize temporal information across frame sequences. These modifications are based on recurrent blocks and fusion of their hidden states.

2. We design a training strategy for the recurrent self-supervised depth prediction task.

3. We conduct experiments and provide results of the impact of our modifications. We provide an ablation study that helps to estimate the effect of each component that we propose. Our experiments show that the proposed methods can be an effective tool for exploiting temporal information for depth estimation in real time.

The code for reproducing experiments is also released accompanying our paper (https://github.com/MariBax/self-supervised-depth-estimation). The paper is structured as

follows. Section 2 gives an overview of the current methods in depth estimation. In Section 3, we provide a theoretical basis for our experiments. Section 4 describes our experimental setup, Section 5 provides their results, and in Section 6 we provide visual examples and their analysis. Section 7 summarizes and concludes the paper.

## RELATED WORK

In this section, we provide an overview of existing depth estimation methods. We begin with supervised models that directly learn to predict depth maps from single RGB images. Next, we address the problem of self-supervised learning and SfM-based methods. Finally, we proceed to existing self-supervised approaches which are most relevant to our work.

### Supervised depth estimation

The idea of supervised depth estimation implies direct learning of a hidden mapping between the input image and its depth. Usually, this approach includes regression-based methods because of the continuous nature of depth maps.

One of the earliest applications of deep learning to this problem (*Eigen, Puhrsch & Fergus, 2014*) introduced a network that consists of two components: one estimates the global scene structure, and the other refines the results using local information. In the stereo-based approach, *Xie, Girshick & Farhadi (2016)* designed a convolution neural network that generates a corresponding right view of a stereo pair by combining information from multiple levels. Disparity distribution for pixels is optimized with supervised photometric loss. *Cao, Wu & Shen (2016)* reformulated the supervised depth estimation task as a pixel-wise classification problem that was solved with a fully convolutional deep residual network.

Supervised methods mostly rely on convolutional architectures. One important difference between methods is that they tend to utilize a large variety of objective functions that define the difference between the ground truth depth map and its prediction. The most straightforward approach is to use $L_1$ or $L_2$ distances. Reverse Huber loss (*Laina et al., 2016b*) is a popular example of a combination of these two distances. To address the scale problem, some studies focused on the loss as a distance between predicted and real depth in log space (*Eigen, Puhrsch & Fergus, 2014*; *Eigen & Fergus, 2014*). *Yin et al. (2019)* argued that most metrics neglect geometric constraints in the three-dimensional space, so they developed a loss that enforces geometrical constraints. Another alternative to pixel-wise metrics is perceptual loss (*Johnson, Alahi & Li, 2016*), which was successfully used in many depth-related tasks (*Makarov, Aliev & Gerasimova, 2017*; *Makarov et al., 2017*; *Makarov, Korinevskaya & Aliev, 2018a*; *Makarov & Korinevskaya, 2019*; *Makarov, Korinevskaya & Aliev, 2018b*; *Makarov, Korinevskaya & Aliev, 2018c*; *Makarov et al., 2019*; *Maslov & Makarov, 2021*). To reduce blurring of estimated depths, a new model with fusion mechanisms was proposed by *Hu et al. (2019)*. It consists of a module that can fuse multi-scale information produced by an encoder, a refine module to fuse decoder outputs and multi-scale fusion modules. In their objective function, the authors combined the difference in gradients and surface normals with balanced Euclidean loss in log space.

State of the art performance in the supervised setting has been achieved by *Fu et al. (2018)*, who proposed the SID policy and ordinal regression loss. Despite the excellent performance of the model, its main disadvantage is a huge number of parameters that leads not only to long training time but also to large inference times in practice. Therefore, a challenge arises to build a lightweight model that is able to process frame sequences in real-time with reasonable quality. Recently, *Maslov & Makarov (2020a)* proposed an architecture that appears to be suitable for real-time robotic applications. The model utilizes temporal information from image sequences with the help of a convolutional gated recurrent unit (ConvGRU) and convolutional long short-term memory (ConvLSTM). The authors showed that these recurrent blocks can lead to improvements in depth estimation accuracy.

## Self-supervised depth estimation

As deep learning models for computer vision become larger and more expressive, the data requirements for training them rapidly grow as well. Creating a labeled dataset for depth estimation by hand or with special hardware is extremely laborious and expensive. One solution would be to use synthetic data (*Nikolenko, 2021*) that has indeed been extensively used for depth estimation. An alternative approach would be to learn from unlabeled monocular sequences or stereo image pairs, as has been done, *e.g.*, in *Garg, G & Reid (2016)* and *Godard, Aodha & Brostow (2016)*. This approach does not utilize ground truth depth maps. One of the first models to perform self-supervised depth estimation (*Garg, G & Reid, 2016*) consists of a convolutional neural network that estimates depth maps from paired stereo images. A predicted depth map is used in the geometrical transformation of the left image into the synthesized right image or vice versa. The loss function here consists of the difference between the synthesized and real images, defined as the structural similarity index measure (*Wang et al., 2004*), and depth regularization terms.

Further studies mostly focused on generalizations of this approach in the monocular setting. Monocular video-based methods usually utilize consecutive frames to simultaneously learn the depth and pose. Given a target image and adjacent frames as input, a model produces the depth for the target image and estimates relative poses between the target image and nearby images. Next, the target view is reconstructed with the predicted depth map and relative poses. The main difference between stereo-based and monocular video-based approaches is that poses for stereo data are known and fixed, while in the monocular setting, they also need to be predicted.

Most state-of-the-art methods in self-supervised depth estimation are based on encoder–decoder architectures. Next, we highlight a few such models that are the most relevant for our study. *Monodepth2 Godard et al. (2019)* has been probably the most influential approach in self-supervised depth estimation because of its simplicity. Its architecture consists of a depth net and a pose net; both nets consist of a ResNet encoder and a decoder. The main contribution of *Monodepth2* is a set of improvements that together lead to superior results in depth estimation. The authors introduced the minimum reprojection loss as a substitute for the average reprojection loss and showed that the new loss is more effective in handling occlusions. In addition, *Godard et al. (2019)* proposed a multi-scale

pyramidal upsampling method and auto-masking loss that reduced visual inconsistency. As a result, the model showed high-quality results on the KITTI benchmark, and since then has become a starting point for many researchers. In this work, we mostly follow *Monodepth2* architecture and propose a set of additional components that can be used to extract information from previous frames.

*Guizilini et al. (2020a)* recently proposed a new architecture called *PackNet-SfM*. Its key feature is the design of packing and unpacking blocks that are considered to preserve the 3D geometrical structure of depth maps. Moreover, the authors presented a new velocity supervision loss that solves the problem of scale ambiguity. Depth scale ambiguity implies that predicted depth maps are usually scaled with predefined constants. In PackNet, camera velocity is estimated and used to compute the loss function that helps reduce the problem with scaling. PackNet has achieved excellent performance on KITTI, but it utilizes heavy 3D convolutions, resulting in a long processing time.

In this work, we intend to investigate the effect of temporal information in depth estimation in the self-supervised setting. Despite the fact that most existing models do not focus on capturing temporal knowledge, there is a number of novel models that try to follow a multi-frame or sequential approach in depth estimation (*Watson et al., 2021*; *Hur & Roth, 2021*; *Kuznietsov, Proesmans & Van Gool, 2021*). In particular, *Watson et al. (2021)* proposed to utilize nearby frames if they are available at test time and designed a model called *ManyDepth*. Its key feature is a cost volume block that helps overcome the scale ambiguity that arises from self-supervised training. Experiments on KITTI and *Cityscapes* show that *ManyDepth* outperforms most existing self-supervised approaches. *Hur & Roth (2021)* focused on the scene flow task where 3D scene flow and depth are estimated jointly in a self-supervised setting. They propose a multi-frame model that uses triplets of frames and recurrent blocks, and it shows state-of-the-art performance for the scene flow task.

## METHODS

The self-supervised setting implies that ground truth depth maps are not available during training. The standard approach here would be to train the model using input image reconstruction as supervision. In this case, the model takes as input a list of monocular images or stereo pairs, where one of the images is called the target image. By estimating the depth in the target image and projecting it to nearby views, we reconstruct the target image and try to minimize the reconstruction error.

In this section, we describe our model in detail. First, we review a common approach to self-supervised depth estimation. We provide details about network structure and objective functions. Next, we describe the components of our network: recurrent block ConvGRU, self-attention layer, and the fusion technique.

### Problem formulation

In the self-supervised monocular setting, we operate with frame sequences extracted from videos. Given these consecutive frames, we predict the target view from the viewpoint of another image. In order to synthesize the target image, we use a hidden variable, the depth

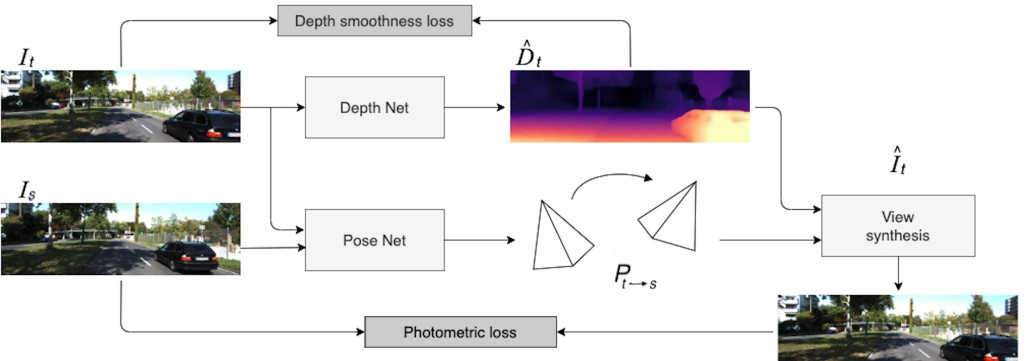

**Figure 1  Self-supervised monocular SfM network architecture.** First, the target image $I_t$ is processed by *Depth Net*. At the same time, target and context frames $(I_t, I_s)$ are passed through *Pose Net*. Then, using 6-DoF relative pose $P_{t\to s}$ and depth map $\hat{D}_t$, the target image $\hat{I}_t$ is reconstructed *via* the inverse warping transformation. The loss consists of two parts: photometric reprojection error between the real target image and the synthesized one and a depth smoothness term. Road images and ground truth depth maps taken from *Menze & Geiger (2015)*.

map, and estimated relative pose between the target image and nearby view. The overall self-supervised pipeline is illustrated in Fig. 1.

Formally, the problem can be described as follows. Suppose that we are given $N$ adjacent images that can be considered as a frame sequence $\{I_1, \ldots, I_N\}$. Usually, one image $I_t$ is chosen as the target image, while other frames $S = \{I_s\}_{s=1,\ s\neq t}^{N}$ are considered as context. This sequence is an input for our model, which consists of two parts:

- the depth network (*Depth Net* in Fig. 1) $f_D : I \to D$ that predicts scale-ambiguous depth $D = f_D(I(p))$ for every pixel $p$ in the input image $I$; using this depth network, we estimate the depth for the target image;
- the pose network (*Pose Net* in Fig. 1) $f_P : (I_t, I_s) \to P_{t\to s}$ that estimates a set of 6-DoF relative poses $P_{t\to s}$, $I_s \in S$ between the target image $I_t$ and nearby context frames; for example, if $N = 3$ then $I_t = I_2$ and $I_s = I_i$, where $i \in \{1, 3\}$.

Given a pair $(I_s, I_t)$ consisting of the target image and source image, we can inversely warp $I_t$ to the source frame $I_s$. In order to do this view synthesis, we need to know an estimated depth map $D_t$ for the target image $I_t$ and the transformation $P_{t\to s}$ from $I_t$ to $I_s$. For a given pixel coordinate $p_t$ from $I_t$ which is co-visible in $I_s$, its corresponding pixel coordinate $p_s$ in $I_s$ can be defined by the inverse warping transformation as follows:

$$p_s \sim K_s[P_{t\to s}]D_t(p_t)K_t^{-1}p_t, \tag{1}$$

where $\sim$ denotes equality in the homogeneous coordinates (*Shen et al., 2019*), and $K_s$ and $K_t$ are intrinsic matrices that are known and fixed. $D_t(p_t)$ is the depth value for a pixel $p_t$, and $[P_{t\to s}]$ is the transformation matrix. Thus, given the depth map and transformation matrix, we can construct a matching function between pixels of the source and target images. This approach can be considered as a geometric projection relationship, and it

helps us synthesize views. The difference between a real image and a synthesized one is a key component in the loss function described below.

## Objective function

In the objective function, we follow *Zhou et al. (2017)*: the overall self-supervised loss function consists of the photometric reprojection error $L_p$ and depth regularization term $L_s$.

Let $I_t$ be a ground truth target image, and let $I_{s \to t}$ be the target image synthesized from the source image $I_s$. Then the basic photometric reprojection error $L_p$ is defined as

$$L_p = \sum_{I_s \in S} \text{pe}(I_t, I_{s \to t}), \tag{2}$$

where the reprojection error is defined as the weighted combination of the structural similarity index measure (SSIM; see *Wang et al. (2004)*) and $L_1$ pixelwise difference:

$$\text{pe}(I_a, I_b) = \alpha \frac{(1 - SSIM(I_a, I_b))}{2} + (1 - \alpha)\|I_a - I_b\|_1. \tag{3}$$

Following *Godard et al. (2019)*, we set $\alpha = 0.85$.

The depth regularization term, first introduced by *Godard, Aodha & Brostow (2016)*, helps to regularize the depth in texture-less low-image gradient regions and is defined as

$$L_s(\hat{D}_t) = |\delta_x \hat{D}_t| e^{-|\delta_x I_t|} + |\delta_y \hat{D}_t| e^{-|\delta_y I_t|}. \tag{4}$$

As we have mentioned above, we calculate the reprojection loss for a set of source images $I_s$. A common strategy used to compute the final loss is to average the reprojection error from multiple sources, *i.e.,* to average the expression in Eq. (2). However, this approach may cause problems with pixels that may be not visible in some source images. Such pixels represent a common problem in the ego-motion study because there is a high probability of encountering objects blocking each other, especially in outdoor scenes. Several works have shown that this problem can be overcome by using masking (*Mahjourian, Wicke & Angelova, 2018*; *Vijayanarasimhan et al., 2017*); however, they do not take into account the recovery of occluded regions. This phenomenon has been carefully studied by *Godard et al. (2019)*, and a simple solution was proposed: the authors replaced averaging with minimum. As a result, the total photometric loss is defined as

$$L_p = \min_{I_s \in S} \text{pe}(I_t, I_{s \to t}). \tag{5}$$

Multi-scale image reconstruction is a common technique in computer vision tasks used to prevent the objective function getting stuck in local minima. Following *Godard et al. (2019)*, we also use multi-scaling in our model. More formally, the photometric loss is defined not just as on the final decoder output but as a combination of loss terms on every scale of the decoder. In this work, we consider 4 scales. *Godard et al. (2019)* revealed that this approach may result in "holes" in low-texture parts of the image, so *Godard et al. (2019)* provided a modification for the multi-scaling approach. They upsample depth maps at low scales of the decoder and then compute the photometric loss with these upscaled maps, so that decoder outputs in the loss computation are upscaled to the original image

resolution. This approach has proven to be very effective in the self-supervised multi-scale depth estimation pipeline.

Self-supervised depth estimation mostly relies on ego-motion. Nevertheless, sometimes motion assumptions may be violated when the camera is in a stationary position but objects move in the scene. As a result, we may have stationary pixels that need to be processed by the model. To overcome these issues, some approaches have been using per-pixel masking (*Zhou et al., 2017*; *Vijayanarasimhan et al., 2017*; *Luo et al., 2018*). The mask is usually incorporated into the loss function, and it can be either predictive, as in *Zhou et al. (2017)*, or computed by some direct algorithm. In this work, we use the auto-masking method proposed in *Godard et al. (2019)*. The idea is to track pixels that remain the same between nearby frames, which implies a static position of the camera. The mask is binary, $\mu \in \{0, 1\}$, where the value 0 corresponds to pixels where the reprojection error of the synthesized image $I_{s \rightarrow t}$ is lower than the error of the original unwrapped image $I_s$:

$$\mu = \left[ \min_{I_s \in S} \mathrm{pe}(I_t, I_{s \rightarrow t}) < \min_{I_s \in S} \mathrm{pe}(I_t, I_s) \right]. \tag{6}$$

*Overall loss function.* The final loss is computed as a sum of the masked photometric reprojection term and a smoothness term. We average this result over each pixel and every image in the batch:

$$L = \mu L_p + L_s. \tag{7}$$

In the next sections, we provide an overview of three components that we incorporate into our model to account for multiple frames at the input: recurrent block ConvGRU, self-attention layer, and fusion. Each of the components is essential to our model as will be shown in the ablation study section.

## Recurrent block ConvGRU

Recurrent neural networks (RNNs) have been widely adopted in the field of computer vision. However, initial RNN models appeared to face difficult training problems, such as exploding or vanishing gradient effects (*Pascanu, Mikolov & Bengio, 2012*). A common way to overcome the vanishing gradient problem is to use long short-term memory units, LSTM (*Hochreiter & Schmidhuber, 1997*; *Gers, Schmidhuber & Cummins, 2000*). LSTMs have become one of the most popular approaches in sequence-to-sequence settings, especially in natural language processing (NLP) tasks, including speech recognition (*Graves, Mohamed & Hinton, 2013*) and machine translation (*Parmar & Devi, 2019*). By definition, an LSTM block operates with one-dimensional inputs and is not naturally suited to image processing. Nevertheless, *Shi et al. (2015)* recently proposed a new architecture called ConvLSTM that extends the basic LSTM construction to two-dimensional inputs.

Over the past few years, some studies have used an alternative recurrent mechanism called ConvGRU (*Zhang et al., 2019*). It was first introduced for the video representation task (*Ballas et al., 2015*) and has proven to be quite effective in subsequent studies (*Siam et al., 2016*; *Maslov & Makarov, 2020a*). ConvGRU has a simpler architecture than ConvLSTM

while preserving the same gated structure. In this work, we use ConvGRU cells since it has fewer parameters, and it has been proven to be effective in capturing temporal information for online depth estimation (*Maslov & Makarov, 2020a*).

Next, we describe the ConvGRU cell as a particular type of RNN network applied to a sequence of 2D objects with an arbitrary length. ConvGRU can be viewed as a modification of the original GRU cell (*Cho et al., 2014*). The main difference between these GRU versions is that ConvGRU utilized convolution operations instead of fully connected units used in the original GRU. Each recurrent unit of the ConvGRU cell is designed to capture dependencies on different time scales. Moreover, due to convolution operation, these recurrent units share their parameters across different spatial locations in the input (*Ballas et al., 2015*). The hidden state $h_t$ of the ConvGRU can be described by the following equations:

$$z_t = \sigma(W_{xz} * x_t + W_{hz} * h_{t-1} + b_z), \tag{8}$$

$$r_t = \sigma(W_{xr} * x_t + W_{hr} * h_{t-1} + b_r), \tag{9}$$

$$\bar{h}_t = \tanh(W_x * x_t + W_h(r_t \circ h_{t-1}) + b_h), \tag{10}$$

$$h_t = (1 - z_t) \circ h_{t-1} + z_t \circ \bar{h}_t, \tag{11}$$

where $*$ denotes convolution, weight matrices $W_{xz}$, $W_{hz}$, $W_{xr}$, $W_{hr}$, $W_x$, $W_h$ are 2D-convolutional kernels, and $b_z$, $b_r$, $b_h$ are the corresponding bias terms. In order to preserve the same spatial size of hidden representations, zero-padding is used in recurrent convolutions. The ConvGRU architecture is illustrated in Fig. 2.

Our main idea of using convGRU unit is similar to *Maslov & Makarov (2020a)*, in which authors used combinations of convolutions and recurrent blocks for temporal information aggregation for supervised depth estimation. We believe that in self-supervised setting temporal consistency is even more essential for robust depth estimation in both, single image and sequential settings. The key component to it is properly chosen self-attention mechanism described next.

## Self-attention

Convolutional neural networks are widely used in many neural network architectures, especially in computer vision. However, a single convolutional layer can only capture dependencies in a small neighborhood. Therefore, alternative approaches based on self-attention are also starting to arise (*Ramachandran et al., 2019*). Self-attention can be considered as an independent layer that can be a good substitute for convolutions. Experiments by *Ramachandran et al. (2019)* show that simply replacing spatial convolutions with self-attention in a ResNet-type architecture leads to higher accuracy in ImageNet classification and COCO object detection. In this work, we use self-attention layers to fuse temporal features.

Next, we describe the self-attention layer used in our architecture in detail; our exposition directly follows the original paper by *Ramachandran et al. (2019)*. Similar to a convolution, given a pixel $x_{ij} \in \mathbb{R}^{d_{in}}$, we first take a neighborhood $N_k(i, j)$ centered on this pixel that extends for $k$ pixels around $x_{ij}$. This region is called a memory block.

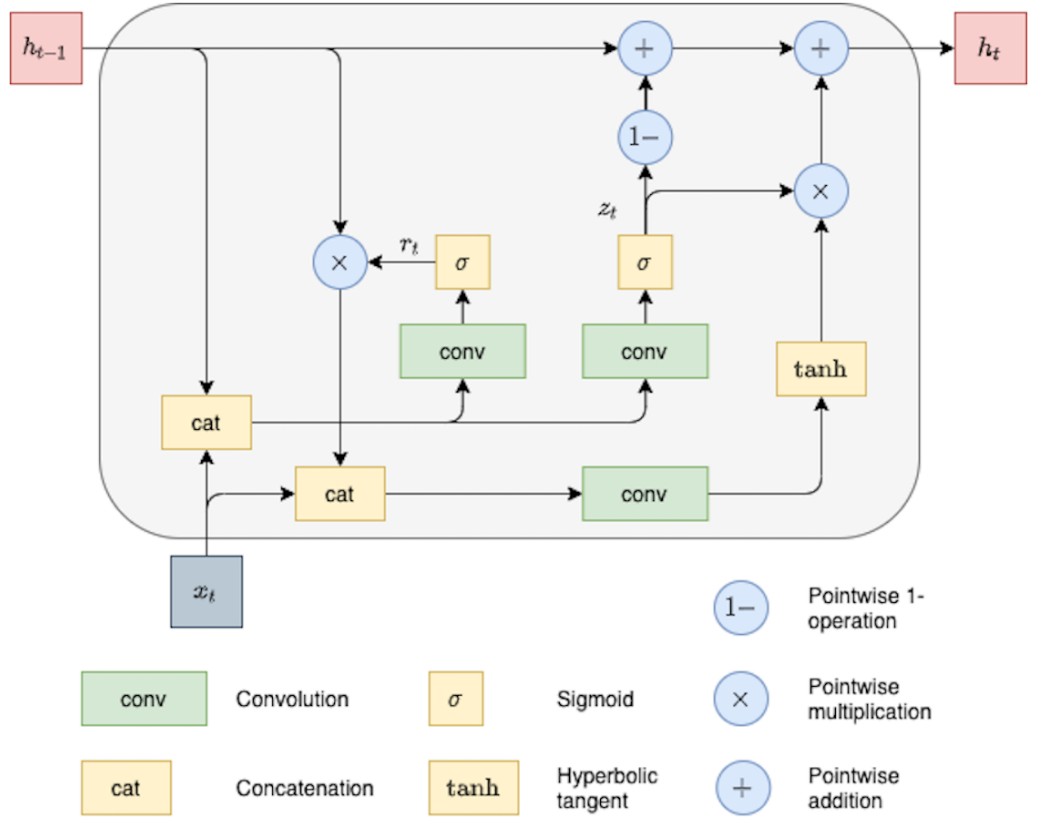

**Figure 2** **ConvGRU block with feature map as input.**

One-headed attention is calculated as follows:

$$y_{ij} = \sum_{a,b \in N_k(i,j)} \text{softmax}_{ab}(q_{ij}^T k_{ab}) v_{ab}, \tag{12}$$

where $y_{ij} \in \mathbb{R}^{d_{out}}$, queries $q_{ij} = W_Q x_{ij}$, keys $k_{ab} = W_K x_{ab}$, values $v_{ab} = W_V x_{ab}$. $W_Q$, $W_K$, $W_V$ $\in \mathbb{R}^{d_{out} \times d_{in}}$ are linear transforms. Similar to convolutions, self-attention aggregates spatial information over a neighborhood region but aggregation is a convex combination of value vectors with weights defined by the softmax operator. This procedure repeats for every pair of indices $(i, j)$. Moreover, it is possible to use multiple attention heads, splitting the pixels $x_{ij}$ depthwise into $N$ groups $x_{ij}^{(n)} \in \mathbb{R}^{d_{in}/N}$, computing the output for each group separately, and then concatenating these outputs.

Next, to aggregate information from different resolutions, frames, and components, we need efficient fusion model taking the best out of convolutional and attention aggregation mechanisms.

## Fusion

Fusion is a technique commonly used to aggregate multimodal features. Multimodality implies that objects have different semantic characteristics. Traditional fusion methods include concatenation or simple averaging. Numerous studies employ fusion for processing

multimodal inputs, and many more complex fusion methods have been proposed over the last decade.

In this work, to fuse temporal features we use residual convolutional units as introduced by *Lin et al. (2016)* and their modification that we call *residual self-attention units*. A residual convolution unit itself is a simplified version of the convolution unit in the original ResNet (*He et al., 2015*). It consists of two spatial convolutions with ReLU activations and batch normalization. It also has a skip connection between the input and the output of convolution layers. In this work, we propose a modification of the residual convolution unit: we replace the spatial convolutional layer with a self-attention layer. We believe that this modification helps capture dependencies from different neighborhoods of the input's representation, potentially resulting in more accurate predictions. Figure 3 shows a schematic overview of the residual self-attention unit (Fig. 3B) and residual convolutional unit (Fig. 3C).

Residual convolutional and self-attention units are used to fuse ConvGRU outputs and hidden states. Fusion modules progressively combine and upscale feature maps in order to generate a fine-grained prediction. Figure 3A presents a schematic overview of our Fusion block.

Finally, we are ready to describe overall model.

## Recurrent self-supervised depth estimation

In this section, we describe the depth network pipeline and illustrate the role of recurrent blocks and fusion units in it. For this study, we have attempted to investigate and propose a modification to existing pipelines that would utilize temporal information across frame sequences. To examine the impact of temporal knowledge, we tested different modifications, and in this section, we describe them in more detail. We have experimented with depth network components, while the pose network remains the same as in previous models (*Godard et al., 2019*; *Guizilini et al., 2020a*; *Zhou et al., 2017*). The pose network consists of the ResNet18 encoder and pose decoder that is presented in Fig. 4.

The key feature of the proposed depth model is the use of two types of temporal information; it is schematically illustrated in Fig. 5. The first type of temporal information is the content of previous frames transferred *via* a recurrent block, *i.e., via* the hidden state. The second type of temporal knowledge comes from the dependencies between different scales of decoder outputs (indicated by green arrows in Fig. 5). The architecture design is inspired by *Sun et al. (2021)* who used similar types of temporal knowledge and connections between them for the real-time 3D scene reconstruction task. In *Sun et al. (2021)*, the recurrent part of the model is based on 3D convolutions, and in this work we simplify their approach and adapt it to our task.

The depth model consists of three consecutive parts. The first part is an encoder–decoder block with skip connections, where the encoder is a ResNet18 and the decoder is a sequence of upsampling convolutional blocks as shown in Fig. 4. The decoder produces outputs on four different scales. We consider two types of decoder outputs: single-channel outputs and multi-channel outputs with $n \in \{16, 32, 64, 128\}$ channels. In the decoder architecture, multi-channel outputs are usually mapped to single-channel ones *via* a convolutional layer,

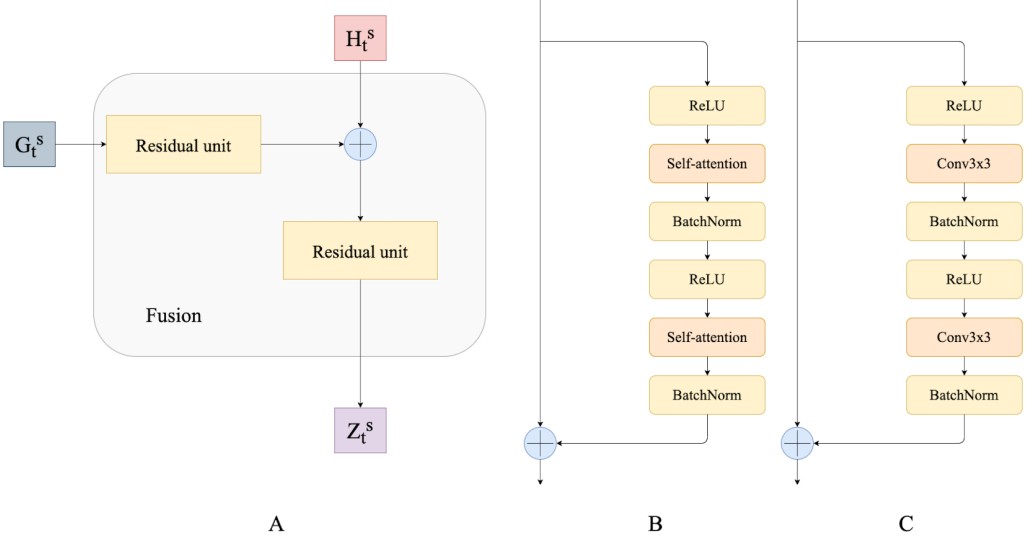

**Figure 3** (A) Fusion block; (B) residual self-attention unit; (C) residual convolutional unit.

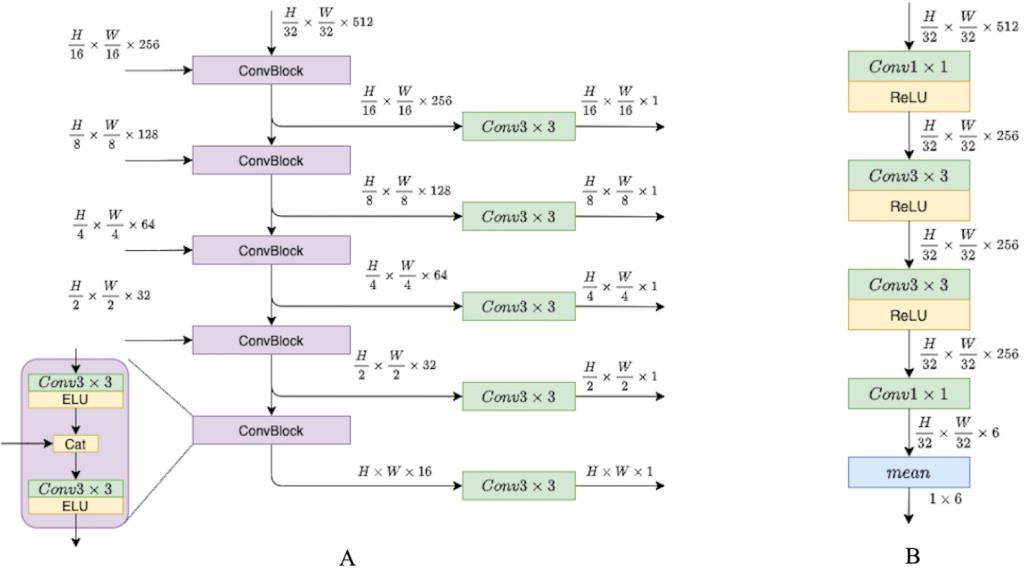

**Figure 4** (A) Depth decoder. (B) Pose decoder. Progressively upsample and fuse depth encoder outputs. Pose encoder outputs pass through convolutions layers with ReLU activations, then flattened to six degrees of freedom translation.

as in *Godard et al. (2019)*. However, in this work we do not complete the depth prediction process at this stage but rather continue to process internal feature representations produced in the decoder. These representations are fed into the second part of our model, recurrent ConvGRU blocks. ConvGRU on a particular scale takes as input a concatenation of the depth output at this scale and an upscaled representation from the previous scale (except

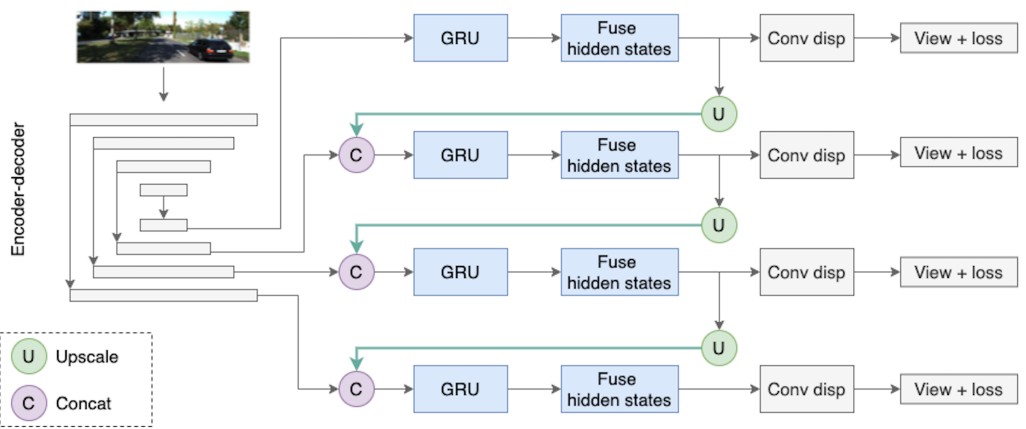

**Figure 5** **The Depth Net pipeline.** First, a target frame is passed through the encoder–decoder architecture. The decoder produces outputs on four different scales, they are passed through recurrent ConvGRU and Fusion blocks, and finally, the depth prediction comes from upscaled maps passed through the convolution disparity layer. Road images and ground truth depth maps taken from *Menze & Geiger (2015)*.

for, obviously, the first scale where ConvGRU receives only the depth output). The third part of the model is the Fusion block that helps aggregate temporal information. The head of the model consists of simple convolution blocks that produce disparity for a target image.

Next, we give a formal description of the full depth estimation pipeline for a given image. Let $X_t$ be an input image, and let $(H_t^1, H_t^2, H_t^3, H_t^4)$ be the hidden states of the recurrent block on four scales produced from previous frames. The encoder–decoder network produces outputs on four scales $s \in \{1, 2, 3, 4\}$. We denote by $F_t^s$ the encoder–decoder output on scale $s$ at time $t$:

$$(F_t^1, F_t^2, F_t^3, F_t^4) = f_{Dec}(f_{Enc}(X_t)). \tag{13}$$

Then, we feed decoder outputs $F_t^s$ and upscaled maps from previous scales into ConvGRU blocks:

$$G_t^s = f_{ConvGRU}^s(concat(F_t^s, U_t^{s-1})), \quad s \in \{2, 3, 4\}, \tag{14}$$

$$G_t^s = f_{ConvGRU}^s(F_t^s), \quad s = 1. \tag{15}$$

Next, we fuse ConvGRU outputs and hidden states from the previous timestamp:

$$Z_t^s = f_{Fusion}^s(G_t^s, H_t^s). \tag{16}$$

Fusion outputs then go to an upscaling block, resulting in $U_t^s$, and to the convolution disparity block that produces final depth map on this scale:

$$U_t^s = f_{Upscale}^s(Z_t^s), \quad s \in \{0, 1, 2\}, \tag{17}$$

$$\hat{D}_t^s = f_{ConvDisp}^s(Z_t^s), \quad s \in \{0, 1, 2, 3\}. \tag{18}$$

For upscaling, we use bilinear interpolation while working with single-channel decoder outputs and the PixelShuffle operation (*Shi et al., 2016*) while working with multi-channel decoder outputs.

## EXPERIMENTS

In this section, we first describe our experimental setup, starting with the dataset, then providing the training details and baseline architectures, and finally detailing the system configuration.

### KITTI Dataset

In order to compare the results with previously proposed monocular self-supervised methods, we evaluate our models on the KITTI 2015 dataset (*Menze & Geiger, 2015*). The data consists of outdoor driving scenes which usually include up to 15 cars and 30 pedestrians. Accurate ground truth depth maps that are needed for further evaluation are also provided.

We use the data split proposed by *Eigen & Fergus (2014)*, with monocular frame sequences. We use the preprocessing technique suggested by *Zhou et al. (2017)* that removes static frames. In order to train the model on sequences, we randomly sample consecutive frames from available scenes. As a result, the training set contains 2685 monocular samples, and the validation set contains 176, where a sample is a sequence of frames. The length of the sequence is set to 10. Following *Godard et al. (2019)*, we use the same camera intrinsics for all images, setting the principal point of the camera to the center of the image and focal length to the average of all focal lengths in KITTI. Training on image sequences is a challenging task for GPU memory limits. Therefore, we had to resize all input images to resolution $194 \times 640$. The same resizing technique was used in previous works (*Godard et al., 2019*; *Guizilini et al., 2020a*; *Zhou et al., 2017*), so comparison with existing models appears to be fair.

### Training details

Working with recurrent neural networks introduces certain complications in the training strategy. We need to choose how to initialize hidden states and how to train the network. The approach to initializing hidden states with zeros is commonly used in many sequence-to-sequence natural language processing tasks, including machine translation (*Parmar & Devi, 2019*). For monocular sequences, we usually use relatively short hidden states because of limited memory resources. Therefore, the impact of the hidden state becomes very important. We follow *Maslov & Makarov (2020a)* and *Patil et al. (2020)* and use two training stages. On the first stage, we train a model for hidden states considering them as parameters, at the second stage we use learned hidden states as initial states for every sequence.

All models are trained for 20 epochs with batch size 2, where one sample in the batch is one sequence of length 10. We use the Adam optimizer with $\beta_1 = 0.9$ and $\beta_2 = 0.999$, learning rate $10^{-4}$ for the first 15 epochs which is then reduced to $10^{-5}$ for the remainder. The smoothness term $\lambda$ is set to 0.001, and SSIM weight $\alpha$ is set to 0.85. Following

*Godard et al. (2019)*, we use pretrained ImageNet weights for the encoder. We also apply the following augmentations: horizontal flips with 50% chance; brightness, contrast, saturation, and hue jitter with ranges of $\pm 0.2$, $\pm 0.2$, $\pm 0.2$, and $\pm 0.1$ respectively.

For the baseline model, we follow *Godard et al. (2019)*: we take the U-net model, where the encoder is a ResNet18 (*He et al., 2015*), and the decoder consists of upconvolutional filters, use the average reprojection term for the loss function, and do not apply scaling to decoder outputs before computing the loss.

### System configuration

All experiments were conducted using a free distribution of Anaconda with Python 3.8. The models were implemented with the Pytorch library (*Paszke et al., 2019*). We used an RTX6000 GPU with 32 GB RAM. The operating system was Ubuntu 18.04.5 LTS.

## RESULTS

In this section, we present the evaluation results for our model and show that it produces competitive results by comparing it with existing self-supervised models. For evaluation metrics, we use standard depth evaluation metrics from *Eigen, Puhrsch & Fergus (2014)*; *Eigen & Fergus (2014)* and the Eigen split setting from *Eigen, Puhrsch & Fergus (2014)*. As has usually been done for KITTI Eigen split evaluation (*Godard et al., 2019*; *Watson et al., 2021*), we clip predicted depths at 80 m, and only evaluate on depth maps with ground truth under 80 m.

### Ablation Study

In order to estimate the effect of each component that we use in our pipeline, we provide an ablation study of the proposed modifications. An ablation study implies changing various components of our model, and its results help to better understand how each component contributes to the overall performance in monocular training. We first compare models with single-image evaluation settings (Table 1). Then we evaluate our models on sequences. The results are illustrated in Table 2. We can see that the baseline, without any of our modifications and contributions from *Godard et al. (2019)*, performs the worst.

There are five different modifications; all models have the ConvGRU block but they differ in the number of channels in decoder outputs, the use of the upscaling block, and the use of the Fusion block. In particular, models #1 and #4 have only ConvGRU layers, other models use the Fusion block and upscaling, and differ only in Fusion block type.

Since our model was trained on sequences, we hypothesize that if we make use of sequence information at test time, we can achieve higher depth prediction accuracy. As a result, we propose two evaluation settings: single-image evaluation and sequential evaluation. Single-image evaluation means that we process each test frame separately. The model, initialized with trained hidden states, takes a single test frame as input and produces a depth map. On the other hand, sequential evaluation implies that we run the model on the previous $n$ frames before producing the depth map for the target test image. We experimented with different values of $n$ and empirically found that 10 frames of history appear to be optimal. During sequential evaluation we "gather" temporal information

**Table 1** **Ablation study for single-image based methods.** Results for different variants of our model with monocular training on KITTI Eigen split. Best results in each group are marked in bold.

| Model | Eval | Dec. output | Upscale & concat | Fuse: ResConv | Fuse: ResSA | RMSE | RMSE log | Abs Rel | Sq Rel | $\delta < 1.25$ | $\delta < 1.25^2$ | $\delta < 1.25^3$ |
|---|---|---|---|---|---|---|---|---|---|---|---|---|
| Baseline | image | 1-ch | – | – | – | 5.512 | 0.223 | 0.140 | 1.610 | 0.852 | 0.946 | 0.973 |
| Monodepth2 | image | 1-ch | – | – | – | 4.862 | **0.193** | **0.115** | 0.903 | **0.877** | **0.959** | 0.981 |
| Model #1 | image | 1-ch | – | – | – | 5.615 | 0.216 | 0.137 | 1.007 | 0.816 | 0.941 | 0.978 |
| Model #2 | image | 1-ch | + | – | + | 5.123 | 0.203 | 0.129 | **0.861** | 0.846 | 0.953 | **0.982** |
| Model #3 | image | 1-ch | + | + | – | 4.982 | 0.199 | 0.120 | 0.877 | 0.861 | 0.955 | 0.981 |
| Model #4 | image | n-ch | – | – | – | 4.998 | 0.201 | 0.123 | 0.891 | 0.857 | 0.956 | 0.981 |
| Model #5 | image | n-ch | + | + | – | **4.851** | 0.200 | 0.124 | 0.875 | 0.859 | 0.956 | 0.980 |

**Table 2** **Ablation study for sequence-based methods.** Results for different variants of our model with monocular training on KITTI Eigen split. Best results in each group are marked in bold.

| Model | Eval | Dec. output | Upscale & concat | Fuse: ResConv | Fuse: ResSA | RMSE | RMSE log | Abs Rel | Sq Rel | $\delta < 1.25$ | $\delta < 1.25^2$ | $\delta < 1.25^3$ |
|---|---|---|---|---|---|---|---|---|---|---|---|---|
| Model #1 | seq | 1-ch | – | – | – | 5.198 | 0.208 | 0.133 | 0.926 | 0.828 | 0.948 | **0.981** |
| Model #2 | seq | 1-ch | + | – | + | **4.811** | **0.199** | 0.126 | 0.910 | 0.858 | **0.957** | **0.981** |
| Model #3 | seq | 1-ch | + | + | – | 4.976 | **0.199** | **0.120** | **0.874** | **0.861** | 0.955 | **0.981** |
| Model #4 | seq | n-ch | – | – | – | 4.963 | **0.199** | 0.122 | 0.879 | 0.859 | 0.956 | **0.981** |
| Model #5 | seq | n-ch | + | + | – | 4.847 | **0.199** | 0.124 | 0.883 | 0.860 | **0.957** | **0.981** |

from previous frames *via* hidden states and utilize this information in the final prediction for the test image. This sequential evaluation approach does reflect real-life scenarios well since in these scenarios the depth estimation component usually does have access to the frame history. As for the KITTI dataset, previous frames are available for all test images except for those at the beginning of a scene; for these frames, we use the maximum available number of previous frames during evaluation.

Despite the fact that most of our models still lose in the final quality in terms of some evaluation metrics to *Monodepth2*, we can make some interesting conclusions regarding the proposed components. For example, all models perform better when they use sequences during evaluation. The upscaling technique also gives a boost to the depth prediction quality. As for the Fusion block, residual self-attention units slightly improve the quality compared to residual convolutional units, as we expected. In contrast, it is hard to say that the use of multi-channel decoder outputs leads to significantly better results. Therefore, this component may be considered redundant, *i.e.,* not contributing to the overall model performance.

In order to choose the best model, we focus on the RMSE metric. Since the errors are squared before averaging, the RMSE score gives a relatively high weight to large errors. Therefore, we consider RMSE to be more reasonable in the depth estimation task, where large errors are particularly undesirable. According to these considerations, we choose model #2 to be the best in our experiments.

**Table 3  Comparison with state-of-the-art image-based methods on KITTI Eigen split.** Input image resolution is 640 × 192. The first column classifies methods as 's' (supervised) or 'u' (self-supervised/unsupervised). Best results in each group are marked as bold.

| Model | Setting | RMSE | RMSE log | Abs Rel | Sq Rel | $\delta < 1.25$ | $\delta < 1.25^2$ | $\delta < 1.25^3$ |
|---|---|---|---|---|---|---|---|---|
| DORN (*Fu et al., 2018*) | s | **2.727** | **0.120** | **0.072** | **0.307** | **0.932** | **0.984** | **0.994** |
| Struct2depth (*Casser et al., 2018*) | u | 5.291 | 0.215 | 0.141 | 1.026 | 0.816 | 0.945 | 0.979 |
| Monodepth2 (*Godard et al., 2019*) | u | 4.862 | 0.193 | 0.115 | 0.903 | 0.877 | 0.959 | 0.981 |
| PackNet SfM (*Guizilini et al., 2020a*) | u | 4.601 | 0.189 | 0.111 | 0.785 | 0.878 | 0.960 | 0.982 |
| *Guizilini et al. (2020b)* | u | **4.381** | **0.178** | **0.102** | **0.698** | **0.896** | **0.964** | **0.984** |
| Ours | u | 4.851 | 0.200 | 0.124 | 0.875 | 0.859 | 0.956 | 0.980 |

Finally, we also provide results of speed tests on RTX6000 GPU. We find that our model runs in real-time with $124 \pm 16$ FPS on resolution $192 \times 640$ compared to DORN with $15 \pm 3$ FPS.

### Results on the KITTI dataset

We compare our best model with previous single-image approaches such as *Godard et al. (2019)*, *Guizilini et al. (2020a)*; *Casser et al. (2018)* and multi-frame sequential approaches such as *Kuznietsov, Proesmans & Van Gool (2021)*, *McCraith et al. (2020)* and *Watson et al. (2021)*. The results are presented in Tables 3 and 4, respectively. We provide scores on the KITTI Eigen split with input resolution $192 \times 640$. In addition, we compare to state-of-the-art supervised models such as *Fu et al. (2018)* and *Maslov & Makarov (2020a)*. We observe that our approach outperforms most single-image self-supervised models in terms of the RMSE score. Our model achieves worse metrics than some sequential approaches, but next, we show that the proposed method better in qualitative comparison.

## DISCUSSION

Our experiments have shown that the model benefits from information collected from previous frames. We assume that incorporation of the recurrent block into a self-supervised training pipeline may provide a potential boost in the results not only in our setting but also in supervised or online supervised training strategies.

We used the KITTI dataset to evaluate our models with standard metrics from *Eigen, Puhrsch & Fergus (2014)* and *Eigen & Fergus (2014)*. In addition to quantitative results, we also provide several visual samples. We present a qualitative comparison with the *Monodepth2* model (*Godard et al., 2019*), since this model served as the basis for our modifications. In Fig. 6 we provide some prediction examples and error maps that are calculated as the absolute difference between the ground truth and predicted depth maps. These visual results show that our model appears to be more accurate in far areas with higher depth variance. We also provide a qualitative comparison with ManyDepth (*Watson et al., 2021*) model, that is the current state-of-the-art self-supervised sequential method. The examples in Fig. 7 illustrate that our model better predicts depth for moving objects like cyclists and cars.

In addition, we noticed that our model produces more accurate depth maps in the neighborhood of objects with non-trivial and triangular shapes, such as people and road

**Table 4  Comparison with state-of-the-art sequence-based methods on KITTI Eigen split.** Input image resolution is $640 \times 192$. The "Setting" column classifies methods as "s-seq" (supervised sequential, i.e., using previous frame information) or "u-seq" (self-supervised sequential). Best results in each group are marked in bold.

| Model | Setting | RMSE | RMSE log | Abs Rel | Sq Rel | $\delta < 1.25$ | $\delta < 1.25^2$ | $\delta < 1.25^3$ |
|---|---|---|---|---|---|---|---|---|
| *Maslov & Makarov (2020a)* | s-seq | **4.104** | **0.170** | **0.101** | **0.707** | **0.887** | **0.964** | **0.986** |
| *Luo et al. (2020)* | u-seq | 4.876 | 0.205 | 0.130 | 2.086 | 0.878 | 0.946 | 0.970 |
| CoMoDA (*Kuznietsov, Proesmans & Van Gool, 2021*) | u-seq | 4.594 | 0.183 | 0.103 | 0.862 | 0.899 | 0.961 | 0.981 |
| *McCraith et al. (2020)* | u-seq | 4.275 | 0.173 | **0.089** | 0.747 | 0.912 | 0.964 | 0.982 |
| ManyDepth (*Watson et al., 2021*) | u-seq | **4.261** | **0.170** | 0.090 | **0.713** | **0.914** | **0.966** | **0.983** |
| Ours | u-seq | 4.811 | 0.199 | 0.126 | 0.910 | 0.858 | 0.957 | 0.981 |

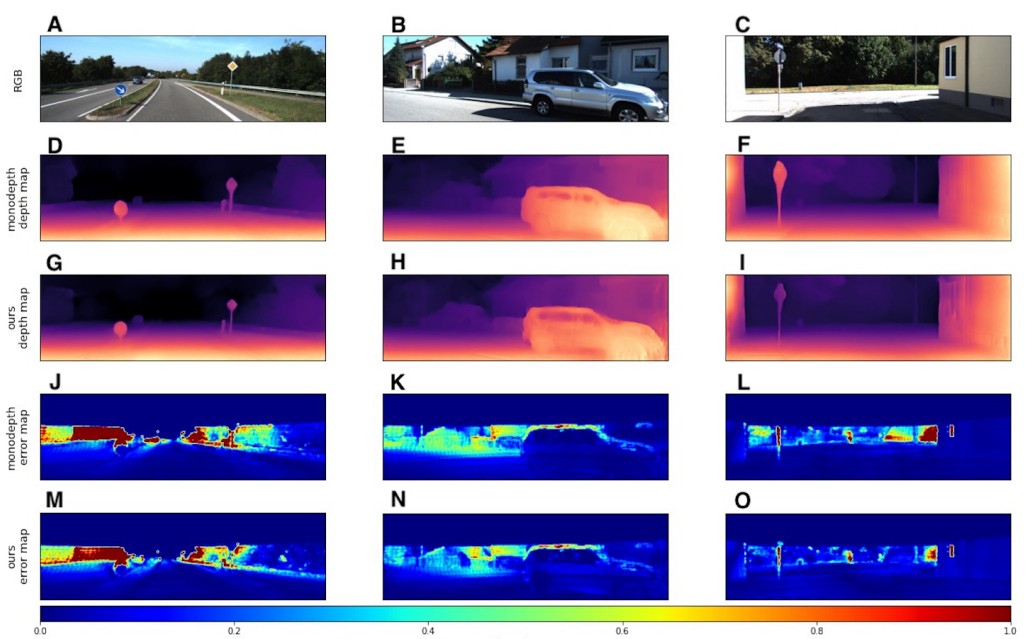

**Figure 6  Qualitative comparison of our model with single-image evaluation setting and Monodepth2** (*Godard et al., 2019*). A, B, C correspond to RGB images. D, E, F correspond to depth output produced by *Monodepth2*. G, H, I correspond to depth output produced by our model. J, K, L correspond to error maps of *Monodepth2*. M, N, O correspond to error maps of our model. Brighter colors on error maps mean higher errors. Road images and ground truth depth maps taken from *Menze & Geiger (2015)*.

signs. We assume that the model performs quite reasonably in these regions because it captures historical information about the frames, resulting in more precise predictions. Qualitative proofs of this evidence are presented in Fig. 8, where we zoom in on the most interesting regions.

Although our model does not achieve state-of-the-art performance in the self-supervised setting, it appears to outperform *Monodepth2 Godard et al. (2019)* and *ManyDepth* (*Watson et al., 2021*) in view synthesis quality. We measured the Frechet Inception Distance(FID) (*Heusel et al., 2017*) of reconstructed test images and saw that our method with FID of 16.863 visually outperforms (*Godard et al., 2019*) and ManyDepth with FID of 16.911 and

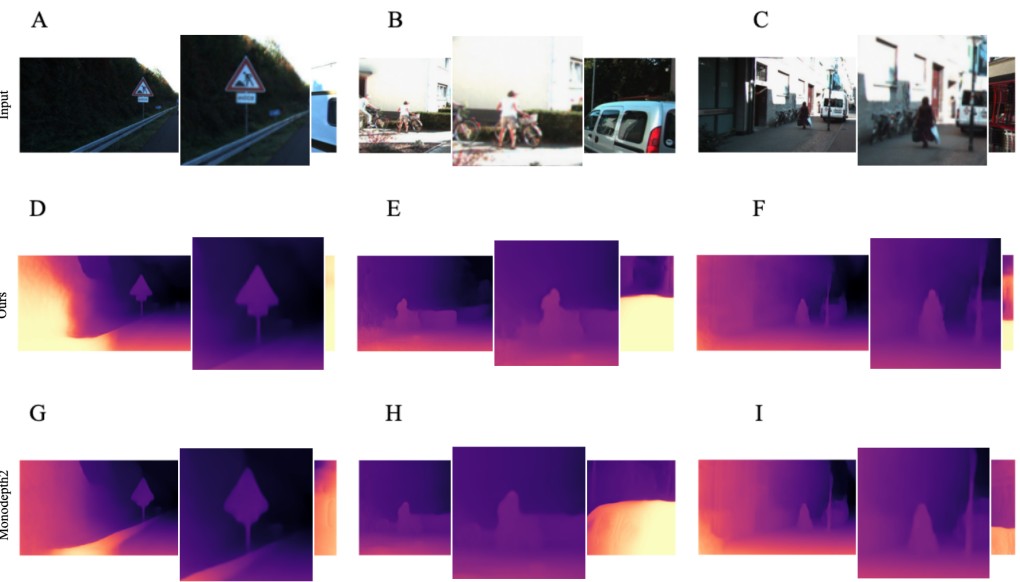

**Figure 7** **Examples where our model with single-image evaluation setting produces more accurate maps than *Monodepth2* (*Godard et al., 2019*) in the neighborhood of objects of non-trivial and angular shapes.** A, B, C correspond to RGB images and their zoomed in regions. D, E, F correspond to the depth output produced by our model. G, H, I correspond to the depth output produced by *Monodepth2*. Road images and ground truth depth maps taken from *Menze & Geiger (2015)*.

16.902 respectively. From the nature of FID, it means that the distribution of features from our model lies more closely to the distribution of features of the ground truth depth maps compared to *Godard et al. (2019)* and ManyDepth, which also leads to more precise boundary localization as shown in the qualitative comparison.

One important open question remains regarding the length of the input sequence. As we have mentioned before, we selected the optimal number empirically. However, we experimented only with short sequences of length up to 12 frames due to memory limits. *Patil et al. (2020)* also tested different sequence lengths and decided that 30 is the optimal value for their setting. In contrast, *Maslov & Makarov (2020a)* used sequences consisting of 10 frames. We assume that this number might be different for different models, and it would be interesting to investigate the effect of the history length and devise optimization strategies to find the optimal value for this parameter for a given model and setting.

## CONCLUSION

In this study, we have presented a self-supervised method that predicts depth from a single image or a sequence of images when available. We provided an ablation study testing different configurations of model components. Experiments show that we have achieved improvements over the basic architecture from both recurrent ConvGRU layer and Fusion module with a self-attention mechanism. We also demonstrate that our method can be applied to real-time. It means that the use of previous frames during training and on test time can provide a significant boost in the accuracy of self-supervised depth estimation.

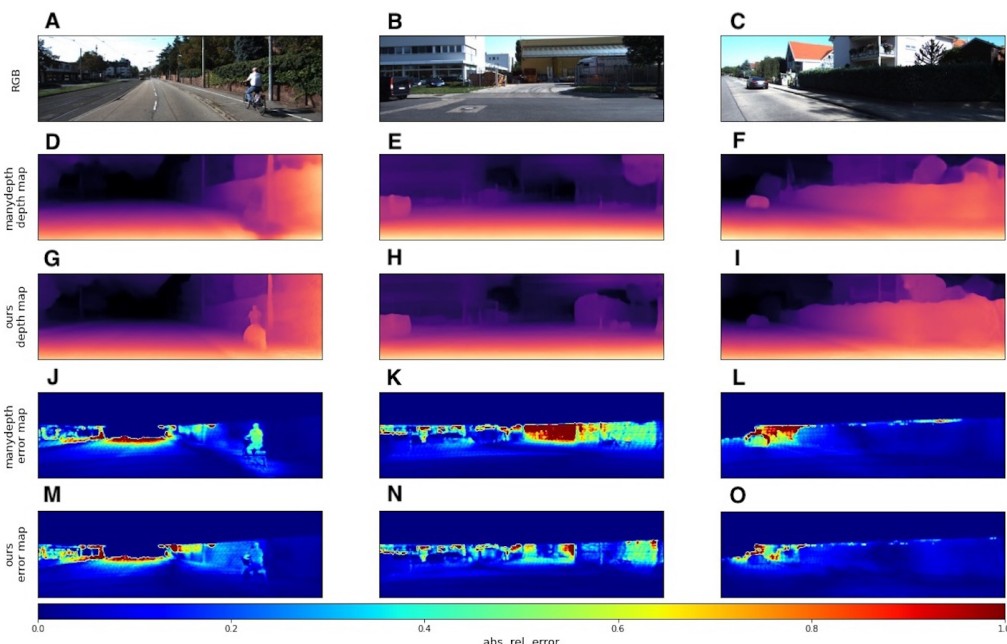

**Figure 8** **Qualitative comparison of our model with sequential evaluation setting and ManyDepth** (*Watson et al., 2021*). A, B, C correspond to RGB images. D, E, F correspond to depth output produced by *ManyDepth*. G, H, I correspond to depth output produced by our model. J, K, L correspond to error maps of *ManyDepth*. M, N, O correspond to error maps of our model. Brighter colors on error maps mean higher errors. Road images and ground truth depth maps taken from *Menze & Geiger (2015)*.

**Future work.** As an interesting direction of future work we suggest the following problems. First, it would be interesting to find an optimal sequence length in both training and evaluation modes. Second, we need to find stabilization techniques for training recurrent networks in a self-supervised pipeline. Third, we expect that our results could be further improved *via* more advanced modifications such as cost volume (*Watson et al., 2021*) or more stable feature-based losses (*Shu et al., 2020*).

# ACKNOWLEDGEMENTS

The authors are grateful to Aleksei Karpov for his invaluable contribution to polishing the article presentation, qualitative comparison, and text.

## Funding

The article was prepared within the framework of the HSE University Basic Research Program. The work of I. Makarov in the Related Work section was prepared in the framework of the federal academic leadership program Priority 2030 of NUST MISIS. The funders had no role in study design, data collection and analysis, decision to publish, or preparation of the manuscript.

## Grant Disclosures

The following grant information was disclosed by the authors:
HSE University Basic Research Program.

## Competing Interests

The authors declare there are no competing interests.

## Author Contributions

- Ilya Makarov and Maria Bakhanova conceived and designed the experiments, performed the experiments, analyzed the data, performed the computation work, prepared figures and/or tables, authored or reviewed drafts of the paper, and approved the final draft.
- Sergey Nikolenko and Olga Gerasimova conceived and designed the experiments, performed the experiments, analyzed the data, prepared figures and/or tables, authored or reviewed drafts of the paper, and approved the final draft.

## Data Availability

The code is available at GitHub: https://github.com/MariBax/self-supervised-depth-estimation.

The data is available at the KITTI Vision Benchmark Suite: http://www.cvlibs.net/datasets/kitti/.

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
