# Peer review of "Self-supervised recurrent depth estimation with attention mechanisms"

_PeerJ Computer Science, doi:10.7717/peerj-cs.865_

## Round 0.1 · original submission · Major Revisions

Reviewers found the paper interesting. However, it needs careful revision according to the reviewers' comments. In particular, better comparison studies (e.g., with/without temporal information) and real-time applicability issue should be included.

Reviewer 1 ·

Basic reporting

The author developed a detph estimation algorithm using self-supervised attention mechanisms. By adopting ConvGRU layers, the networks can capture temporal information even better. Also, the proposed algorithm is precisely compared with the state-of-the-art algorithms in various aspects.

In the introduction section, the author organized the related works clearly, which is quite reasonable to me.

Experimental design

1. Table 1 contains too much infomation. Some of them are useful to some readers but it appears to weaken the arguments. The author needs to simplified the table for more clarity.

2. If the author can upload the codes in the github, then the previous comments can be removed. The author can upload detailed information in the github rather than describing many details of the networks in the paper.

Validity of the findings

1. The table 3 is not clearly explained. The author needs to specify which algorithms are based on single images.

2. The proposed method shows worse performance than some single image baselines even though it uses much information. Then, what is the advantage of using the proposed algorithm beyond the performance?

3. How long does it take in a single feedforward pass? The author assumes that the proposed method has a potential to be used in autonomous driving environments. Then the algorithm should be ran in real-time, otherwise the paper fails to show its validity.

4. What is the performance of a depth estimation algorithm without using temporal information? The author needs to show the comparison results in the ablation study as the author states that using temporal information is the key of the algorithm.

5. In the conclusion section, the author compares the performance with Monodepth2, which seems to me that it is not on the right section. The author can move the statement to the discussion section for clarity.

6. In the conclusion section, the author mentioned Frechet Inception Distance (FID) as an evaluation metric. What is the meaning of it? Can it be a valid metric for comparision? If it is, what is the purpose of selecting FID as a metric?

Additional comments

This paper applies convGRU with self-attention mechansim, which is quite novel to me. However, the result sections fails to illustrate the validity of the proposed framework as the advantage of the algorithm is not clearly described.

Reviewer 2 ·

Basic reporting

The writing overall is comprehendible but the authors can improve upon by better conforming to professional standards of expressions. There are a few expressions throughout the manuscript that do not seem suitable for academic writing; "a bunch of" would be one example. The authors also have a mixed usage of British and American English, such as 'regularisation' and 'regularization', or color. I suggest that the authors proofread the manuscript and address such issues prior to the final submission.

The literature review and introduction seem to provide sufficient background and context regarding the subject. While most figures are able to provide enough context to the readers and are placed well, the figure describing the ConvGRU block (Fig. 2), appears to be identical to that of a conventional GRU block. The difference between a ConvGRU block and a conventional GRU block are adequately described in lines 263 through 269, as well as Equations 8 through 11, hence Fig. 2 may be unnecessary as GRU blocks are typically well-known throughout the field.

Experimental design

The problem formulation seems to be well defined, yet some sections lack to provide enough logical reasoning behind the selection of each method.

While each of the methods separately are described with sufficient detail, it would be clearer for the readers if the authors outline the overall methodology, in terms of presentation of the contents in the methodology section. This will also help with the transition from each subsection to another,

The experimental details are well described and comprehensive enough for replication.

Validity of the findings

The ablation study seems to provide a detailed analysis and justification for the models/methods used in this work. However, the authors should better emphasize the contributions of this paper as the results generally seem to be less convincing to the potential audience. Although the results do suggest that the usage of temporal information can enhance the performance as the authors claim, the proposed methodology performs worse compared to most of the 'u-seq' or models (which are in the same category), and the qualitative comparison of the view synthesis quality was done with a model from a different category. Despite the relative weakness in terms of RMSE compared to other models within the same category, conducting a qualitative comparison with one of the state-of-the-art methods and showing how similarly the proposed method can perform would be help potential readers to be more persuaded. Another suggestion would be to point out and highlight any other advantages the proposed methodology has over the other methods within the same category to compensate for the weakness in performance.

Additionally, for Fig. 5, what are the ranges of the error maps? While the proposed methodology does seem to have less intensive values in the error map, it would be worth indicating the levels by having a colorbar next the the plots in J through O.

---

## Round 0.2 · Minor Revisions

Two reviewers agreed that you revised the paper accordingly. However, there are still some minor issues to be dealt with. Please read reviewer comments carefully and address all of them.

Reviewer 1 ·

Basic reporting

The responses of the author have removed my questions. It could give a better understanding to potential readers if some of the contents are reinforced.

Experimental design

no comment

Validity of the findings

1. The author mentioned that 'supervised learning methods are very heady and self-supervised learning methods are light.' This could be seen as a hasty generalization mistakes as the computational burden varies with the structure and design of networks. If there are any references that support your claim, then adding them in the introduction section could eliminate the concerns.

2. The captions (or y-axis) of the figures 6 and 8 are mislabeled. The explaination of those two figures are not matched with the one from the captions. The author should check the 'D, E, F' and 'G, H, I' parts.

3. The computational speed of the proposed algorithm is quite astonishing and can be applicable to real-time, which could be considered as an advantage of using your algorithm. It would be much better if this contents are emphaized in the manuscript.

Additional comments

The author mentioned that the codes are included in the original paper but It would be better to add the link of the code in the manuscript so that the potential readers can refer your methods easily.

Reviewer 2 ·

Basic reporting

The writing overall has been improved, though the authors could proofread the manuscript carefully prior to the final submission. For example in the first sentence of Section 4, the authors have written "we first our experimental setup", and it seems like there could be a missing verb. Again, it is recommended that the authors review the manuscript for any grammatical or spelling errors.

Experimental design

No comment

Validity of the findings

1. Though the authors claim the contributions of the proposed methodology to be of utilizing new network structures and having new training strategies, how are these finding reflected in terms of performance? As the authors describe in Section 5.2, the proposed model performs worse than most of the multi-frame and sequential approaches, possibly weakening the authors' claim that incorporating temporal information could be beneficial. Novelty alone may not be sufficient for the justification and necessity of this method, and the fact that the proposed method perhaps underperforms most of the state-of-the-art methods would be less convincing to potential readers. Thus it is highly recommended that the authors either (1) highlight any advantages the proposed method holds over other existing methods (why this method might be useful despite worse performance) or (2) enhance the performance of the model such that acceptable performance is achieved.

2. The qualitative comparison with Monodepth 2 does not help in terms of convincing potential readers the usage of the proposed method. One of the main contributions that the authors claim are that sequential/temporal information can be utilized; it would logically make more sense to have a comparison between another sequential model.

---

## Round 0.3 · accepted · Accept

You paper is now acceptable.

Reviewer 1 ·

Basic reporting

No comment

Experimental design

No comment

Validity of the findings

No comment

Additional comments

No comment

Reviewer 2 ·

Basic reporting

The authors have seemed to remove all typos and grammatical errors.

Experimental design

No comment

Validity of the findings

The authors have seemed to address previous comments and questions in the revised manuscript and rebuttal form.